# Ecological-level factors associated with tuberculosis incidence and mortality: A systematic review and meta-analysis

Alemneh Mekuriaw Liyew[1,2,3]*, Archie C. A. Clements[3,4], Temesgen Yihunie Akalu[1,2,3], Beth Gilmour[2,3], Kefyalew Addis Alene[2,3]

1 Institute of Public Health, College of Medicine and Health Sciences, University of Gondar, Gondar, Ethiopia, 2 Faculty of Health Sciences, School of Population Health, Curtin University, Perth, Australia, 3 Geospatial and Tuberculosis Research Team, Telethon Kids Institute, Nedlands, Australia, 4 Research and Enterprise, School of Biological Sciences, Queen's University Belfast, Belfast, United Kingdom

* a.liyew@postgrad.curtin.edu.au

**Data Availability Statement:** All relevant data are within the manuscript and its Supporting Information files.

## Abstract

Globally, tuberculosis (TB) is the leading infectious cause of morbidity and mortality, with the risk of infection affected by both individual and ecological-level factors. While systematic reviews on individual-level factors exist, there are currently limited studies examining ecological-level factors associated with TB incidence and mortality. This study was conducted to identify ecological factors associated with TB incidence and mortality. A systematic search for analytical studies reporting ecological factors associated with TB incidence or mortality was conducted across electronic databases such as PubMed, Embase, Scopus, and Web of Science, from each database's inception to October 30, 2023. A narrative synthesis of evidence on factors associated with TB incidence and mortality from all included studies, alongside random-effects meta-analysis where applicable, estimated the effects of each factor on TB incidence. A total of 52 articles were included in the analysis, and one study analysed two outcomes, giving 53 studies. Narrative synthesis revealed predominantly positive associations between TB incidence and factors such as temperature (10/18 studies), precipitation (4/6), nitrogen dioxide (6/9), poverty (4/4), immigrant population (3/4), urban population (3/8), and male population (2/4). Conversely, air pressure (3/5), sunshine duration (3/8), altitude (2/4), gross domestic product (4/9), wealth index (2/8), and TB treatment success rate (2/2) mostly showed negative associations. Particulate matter (1/1), social deprivation (1/1), and population density (1/1) were positively associated with TB mortality, while household income (2/2) exhibited a negative association. In the meta-analysis, higher relative humidity (%) (relative risk (RR) = 1.45, 95%CI:1.12, 1.77), greater rainfall (mm) (RR = 1.56, 95%CI: 1.11, 2.02), elevated sulphur dioxide (µg m–3) (RR = 1.04, 95% CI:1.01, 1.08), increased fine particulate matter concentration (PM2.5) (µg/ m$^3$) (RR = 1.33, 95% CI: 1.18, 1.49), and higher population density (people/km$^2$) (RR = 1.01,95%CI:1.01–1.02) were associated with increased TB incidence. Conversely, higher average wind speed (m/s) (RR = 0.89, 95%CI: 0.82,0.96) was associated with decreased TB incidence. TB incidence and mortality rates were significantly associated with various climatic, socioeconomic, and air quality-related factors. Intersectoral collaboration across health, environment, housing, social welfare and

---

**Funding:** This work was supported by the Australian National Health and Medical Research Council (NHMRC) through an Emerging Leadership Investigator Grant APP1196549 (KAA) and Curtin University strategic scholarship (AML). The funders had no role in study design, data collection and analysis, decision to publish, or preparation of the manuscript.

**Competing interests:** The authors have declared that no competing interests exist.

economic sectors is imperative for developing integrated approaches that address the risk factors associated with TB incidence and mortality.

## Introduction

Tuberculosis (TB) is a communicable disease caused by the bacillus *Mycobacterium tuberculosis*, that mainly affects the lungs (pulmonary TB) but which also has the potential to disseminate to other body sites (extra-pulmonary TB) [1]. An estimated 10.6 million people fell ill with TB in 2022, an infection that is one of the deadliest in the world, contributing to the loss of more than 1.3 million lives per year [2]. Thirty countries, mostly located in Asia and Africa, are designated as high-TB-burden countries, having high numbers of cases and deaths [2].

The World Health Organization (WHO) has set a target, relative to a 2015 baseline, of a 95% reduction in TB mortality, a 90% reduction in TB incidence and zero TB catastrophic costs, to be achieved by 2035 [3]. Although there has been a gradual decline in TB incidence, an increased number of TB-related deaths have been reported in recent years, partly due to the impact of the Coronavirus Disease 2019 (COVID-19) pandemic, which might have initiated a reversal in global progress in providing essential TB services including diagnosis and treatment [4]. The emergence and spread of drug-resistant TB (DR-TB) also presents a significant threat to global TB prevention and control efforts [5].

TB infection, transmission, and progression to active disease are related to several risk factors that operate at individual and ecological levels [6]. For instance, the existing literature shows that demographic factors such as age, sex, and income [7, 8] as well as clinical factors such as HIV infection [9, 10], poor nutrition [10, 11], and diabetes [11] are individual-level factors associated with TB infection and disease activation.

In addition, ecological factors, factors measured at population level, have been significantly associated with the incidence of TB. These ecological factors capture the social, health care and physical environmental influences that affect TB epidemiology. Such factors include socio-economic conditions [12–14], migration rate [12, 13], gross domestic product and national income level per capita [12], education and employment rate, [13], density of health service provision to the community [14], household crowding [13, 15], and climatic factors, including average temperature, humidity, wind speed, and sunshine duration [16–18]. Although several analytic studies demonstrate significant associations of these ecological-level factors with TB incidence [14, 19–21] and mortality [22–24], the direction of association for each factor has often been inconsistent and conflicting. For example, inconsistent findings were reported on the association of TB incidence with population density [25–27], economic status [28, 29], and wind speed [30, 31]. A comprehensive systematic review and meta-analysis is therefore required to clarify the inconsistencies in existing research and provide synthesised information for decision-making to support the End TB Strategy goals of reducing TB incidence and mortality. By thorough review of ecological factors, we can lay the foundation for developing targeted interventions that address specific risk factors contributing to TB transmission and mortality within communities. Thus, this study aims to qualitatively synthesize and quantitatively analyse the ecological-level drivers of TB incidence and mortality.

## Methods

This study was designed and reported according to the Preferred Reporting Items for Systematic Reviews and Meta-Analysis (PRISMA) guidelines [32]. The protocol for this systematic review was registered in PROSPERO (CRD42023396072).

## Study identification

We conducted a systematic search across multiple databases, including PubMed, SCOPUS, Web of Science, and Embase. This search aimed to identify studies that specifically investigated the association between ecological factors, namely, climatic variables, environmental factors, and indicators measured at an aggregate population level and TB incidence and mortality. We searched each database from inception to April 15, 2023, and undertook an update on October 30, 2023. The search was done without applying restrictions on the type of population (adult versus paediatric), geographic location, language, or date of publication using search terms that combined key concepts such as TB, incidence, prevalence, mortality, and ecological factors. For instance, the following search strategy was applied in PubMed database. "((tuberculosis OR TB OR mycobacterium tuberculosis OR MTB OR pulmonary tuberculosis OR PTB OR extrapulmonary tuberculosis OR EPTB) AND (incidence OR incidence rate OR cumulative incidence OR prevalence OR proportion OR mortality OR mortality rate)) AND (ecologic OR ecological OR ecologic level OR ecological factor OR ecologic factors OR environment OR environmental factors OR climate change OR climatic OR climatic factors OR meteorological factors OR air pollution OR air pollutants OR air quality-related factors) Filters: Humans)". The specific search strategy was designed in the context of individual databases and is described in the S1 Table. Additional papers were identified by hand searching the reference lists of included articles.

## Study screening and inclusion

All articles identified in the databases were imported to EndNote 7.0. After removing duplicates, the articles were exported to Rayyan for screening [33]. Two investigators (AML and TYA) independently screened the titles and abstracts of studies and reviewed full-text articles for inclusion, using predefined eligibility criteria. Any differences were resolved through discussion with a third reviewer (KAA).

Peer-reviewed articles that investigated different types of ecological-level factors including climatic, environmental, and aggregated population-level factors were included. Conference abstracts and letters to editors without adequate information, articles in languages other than English, systematic reviews, and those with insufficient information on the primary outcomes of interest (i.e., TB incidence and mortality) were excluded. Studies conducted on non-tuberculous mycobacteria, animal diseases, and population immunological profiles were also excluded. Full-text articles without sufficient information on the ecological factors examined were also excluded. For excluded studies, the reasons are provided as S6 Table.

## Data extraction

Data extraction was undertaken using a pre-tested checklist and stored in a Microsoft Excel 2016 spreadsheet (Microsoft Corporation, Redmond, Washington, USA). Two reviewers AML and TYA were involved in the data extraction and any disagreements were resolved by consensus.

The following information was extracted from each paper: **I)** study characteristics such as the name of the first author, year of publication, country of the study, study setting, study design, sample size, study aims, data type (notification, prevalence, incidence), a geographic unit of analysis (province, district, village etc), analytic methods (conditional autoregressive modeling, non-bayesian regression models, geographically weighted regression models, Poisson regression model, logistic regression models and other regression methods) and key conclusions; **II)** participants' characteristics such as study population, age and sex; **III)** exposure characteristics such as all ecological factors, including those that weren't significantly

associated with the outcome, and measures of association (relative risk, odds ratio, or beta coefficient) (S1 Data).

## Statistical analysis

In this study, when a meta-analysis of effect estimates was not deemed appropriate due to incompatible effect sizes reported and different variable definitions used across studies, a narrative synthesis method called Synthesis Without Meta-analysis (SWiM), as outlined in the PRISMA extension guideline [34], was utilized. Firstly, studies were categorized based on the type of ecological factors they examined, such as climatic, air quality-related, and socioeconomic factors. Since pooling effect sizes via meta-analysis wasn't feasible due to the inconsistencies in reported effect sizes and variations in variable definitions across studies, we employed a narrative synthesis method called vote counting based on the direction of effect [34]. This approach involved tallying the direction of association reported in each study for a particular ecological factor. For each ecological factor, the direction of association (positive, negative, or neutral) was noted based on the effect sizes reported in the individual studies. Summary tables and graphs were then created to present the overall direction of association for each factor across the included studies. Additionally, detailed narration was provided in the text, describing the number of studies that reported a given direction of association among all the studies that have investigated a particular factor. A random effects meta-analysis was conducted when applicable to quantify associations between ecological-level factors and TB incidence. Random effects meta-analysis is particularly valuable when synthesizing results from studies with varying underlying populations or methodologies. Random effects meta-analysis accounts for both within-study and between-study variability, and enhances the robustness of the conclusions drawn [35]. Heterogeneity between studies was assessed visually by forest plots and quantitatively by the index of heterogeneity squared ($I^2$) statistics with 95% confidence interval. An $I^2$ value greater than 75% was considered as evidence of substantial heterogeneity [36]. The analysis was conducted using STATA (version 17.0).

## Quality assessment

Two authors (AML and TYA) independently assessed the risk of bias. An adapted quality assessment tool designed for ecological studies that evaluates three domains of a given study (i.e. design, statistical methodology, and reporting quality) was used [37]. The tool has 15 items with a maximum score of 21 (i.e., 12 for study design; 6 for statistical methodology, and 3 for quality of reporting) (S5 Table). Studies that scored below 75% were declared to have low methodological quality. The risk of publication bias was assessed statistically by conducting Egger's regression test where a significant p-value ($<0.05$) indicates publication bias.

# Results

In total, 7218 articles were identified from the database search and other sources. After removing duplicates, 5552 unique articles were screened by title and abstract and 128 articles were eligible for full-text review. After a full-text review, 52 [19–24, 29–31, 38–80] articles were included in the systematic review, with one publication investigating two outcomes i.e., TB incidence and mortality, giving 53 studies (Fig 1).

## Characteristics of included studies

Table 1 summarizes the main characteristics of the included studies. All included articles were published between 2003 and 2023. The studies were conducted in 17 different countries mostly

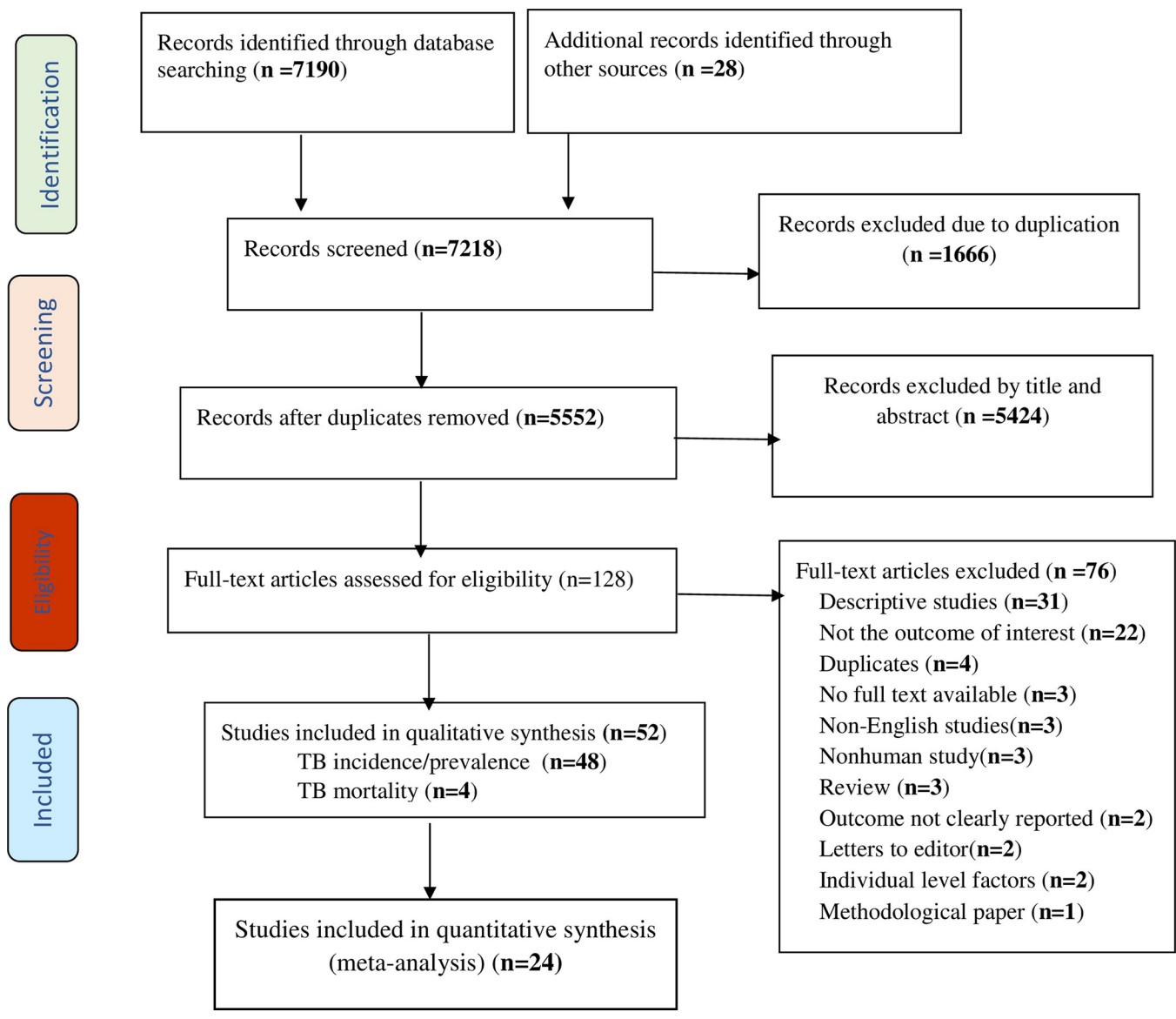

**Fig 1. Flow chart illustrating systematic review process.**

located in Asia and Africa. Most studies (n = 42) analysed all types of TB, without distinction between pulmonary TB and extrapulmonary TB. Few studies included pulmonary TB only (n = 10). The association between ecological-level factors and TB incidence was reported in 48 studies. Few studies reported the association of ecological-level factors with TB mortality rates (n = 4). Most studies used retrospective TB notification data including TB surveillance and TB program data (Table 1 and S2 Table).

Of the included studies, 38 studies were geospatial studies at various units of geographic scale ranging from small villages to larger regions. The commonest unit of analysis was district (n = 10) followed by province (n = 7), with other spatial scales including municipality (n = 4), county (n = 4), census sectors (n = 3), prefectures (n = 3), clusters (n = 1), human development units (n = 1), local areas (n = 1), villages (n = 1), regions of surveillance (n = 1), and subdistricts (n = 1). Fourteen studies used conventional regression models including generalized linear,

**Table 1. The characteristics of 52 studies included in the systematic review and meta-analysis.**

| First author (Year) | Primary outcome | Analytic method used | Ecological level factors investigated | Direction of association |
|---|---|---|---|---|
| Alene (2019) * | Incidence | Bayesian Poisson regression model | Poor healthcare access and good knowledge about TB | Negative |
| | | | Low wealth index and proximity to national borders | Not significant |
| Alene (2022) | Prevalence | Bayesian logistic regression model | Population density | Positive |
| | | | Temperature, altitude, travel time to city, distance to water body, distance to health facility, Precipitation | Not significant |
| | | | Poor knowledge towards TB and higher mean temperature | Positive |
| Alene (2017) * | Incidence | Bayesian spatial Poisson regression models | Proportion of illiterate, use of firewood for cooking (%), high temperature (˚C) and rainfall (mm) | Not significant |
| | | | Proportion of new internal migrant and urban population (%) | Positive |
| Alene (2017) * | Incidence | Spatial Poisson regression analysis | Population density and Migrant population (%) | Not significant |
| | | | Economically inactive population (%), male population (%), urban residence (%) | Positive |
| Alene (2021) * | Incidence | Bayesian spatial Poisson regression models | Proportion of males in a county, percentage of urban residents, birth rate, Gross domestic product | Not significant |
| | | | Low prevalence of contraceptive use, and low sunshine hours | Negative |
| Alves (2020) | Mortality | Spatiotemporal Bayesian models | Aging rate, average household income per capita of the poor, average household income per capita of vulnerable to poverty | Not significant |
| | | | Increase in average per capita household income | Negative |
| Amsalu (2019) | Incidence | Bayesian conditional autoregressive model | Population density, rainfalls, gross domestic product | Positive |
| | | | Air temperature, average wind speed | Protective |
| | | | Average humidity | Not significant |
| Arcoverde (2018) | Mortality | Global and Local Bivariate Moran I | Density of resident and proportion of people of brown skin colour | Positive |
| | | | High income | Negative |
| Bie (2021) * | Incidence | Spatial-temporal Poisson model | Average temperature, average air pressure | Negative |
| | | | Average relative humidity, per capita GDP, average sunshine duration and average precipitation | Positive |
| | | | Average wind speed | Not significant |
| Cao (2016) * | Incidence | Spatial temporal under Bayesian framework | Average temperature, average rainfalls, and Average air pressure | Positive |
| | | | Average wind speed | Negative |
| | | | Average humidity | Not significant |
| Carrasco-Escobar (2020) * | Incidence | Bayesian negative binomial regression model | Particulate matter (air quality) and poverty level | Positive |
| Chen (2023) | Incidence | Conditional autoregressive model | SO2, proportion of people engaged in agriculture | Positive |
| | | | PM10, population density, per capita GDP, low income | Not significant |
| | | | NO2, urbanization rate | Negative |
| Couceiro (2011) | Incidence | Multivariate regression models | High HIV/AIDS rates, non-standard accommodation, overcrowded housing, unemployment | Positive |
| Cui (2019) | Incidence | Spatial panel models with spatial lag and spatial error | Altitude, average temperature, average humidity, forest cover, total GDP, TB control fund, health fund, number of hospitals, number of health facilities, number of doctors, HIV/AIDS prevalence | Not significant |
| | | | Annual rainfall, per capita GDP, treatment success rate of TB, participation rate of rural areas insurance | Negative |
| da Roza (2012) | Incidence | Bayesian Poisson regression | Income level, education | Positive |
| | | | Social vulnerability | Negative |
| de Abreu (2016) * | Incidence | Multivariate logistic analysis | Income level, percentage literacy | Not significant |
| | | | High Population density | Positive |
| Feske (2011) | Incidence | GWR | Poverty, age, black race, and foreign birth | Varies |

(*Continued*)

**Table 1.** (Continued)

| First author (Year) | Primary outcome | Analytic method used | Ecological level factors investigated | Direction of association |
|---|---|---|---|---|
| Gelaw (2019) | Incidence | Regression Tree analysis | Urban residence (%), male population (%), internal migrants (%), cooking with charcoal, use dung for cooking (%), Room crowding | Positive |
| | | | Population density, cooking with wood, PLHIV enrolled in ART | Not significant |
| | | | PLHIV per 1000 population | Negative |
| Sousa (2022) | Incidence | GWR | Population with piped water (%), Population density | Positive |
| GUO (2017) | Incidence | Spatial Poisson regression | GDP per capita, temperature, wind speed, atmospheric pressure | Negative |
| | | | Proportion of the elderly (%), relative humidity (%), | Positive |
| Mohidem (2021) | Incidence | Geographically weighted regression (GWR) | Income status, nationality, residency, NO2, SO2, particulate matter, rainfall, temperature, atmospheric pressure | Differ with location |
| He (2020) | Incidence | Spatial autoregressive models | Proportion of internal emigrants (%), UR urbanization rate (%) | Negative |
| | | | Population density, internal immigrants (%), per capita GDP | Positive |
| | | | proportion of educated population | Not significant |
| Im (2021) | Incidence | Spatial regression model | Population composition, population growth rate, Health insurance | Negative |
| | | | Number of people per medical personnel, SO2, mean temperature | Not significant |
| Li (2022) | Incidence | Geographically weighted regression | Temperature, precipitation, PM10 | Mostly Positive |
| | | | Humidity, SO2 and NO2 | Mostly negative |
| | | | Ozone (O3) | Mostly positive |
| Li (2014) | Prevalence | Partial least squares path modelling and GWR | TB investment, higher elevation, higher humidity, higher temperature, more precipitation, and little sunshine exposure | Positive |
| | | | TB service, health level | Not significant |
| | | | Health investment, economic level, air quality | Negative |
| Liu (2020) * | Incidence | Spatial-temporal Poisson | Particulate matter (PM10) and NO2 | Not significant |
| | | | Sulphur dioxide (SO2) | Positive |
| Munch (2003) | Incidence | Spatial Poisson regression | Crowding, unemployment (%) and alcohol-drinking places (%) | Positive |
| Rao (2016) | Incidence | A spatial panel model | Monthly average temperature, monthly average wind speed | Positive |
| | | | Monthly precipitation | Negative |
| | | | Monthly total sunshine hours | Not significant |
| Rasam (2019) | Incidence | Logistics regression model | Urbanization, distance to factory, Socioeconomic status, distance to healthcare centres, number of populations in a house | Positive |
| Sadeq (2018) | Incidence | OLS | Annual Rainfall, prefecture versus province, HIV infection | Positive |
| Ghadimi (2020) * | Incidence | Spatial logistic regression | Mean annual humidity, urban land cover | Positive |
| | | | Slope | Negative |
| | | | mean annual temperature, mean annual rainfall, Elevation | Not significant |
| Sohn (2019) [a] * | Incidence | Spatial analysis with Poisson regression | Deprivation index, O3, CO | Not significant |
| | | | SO2, Total number of stores and markets, Diabetes (%) | Positive |
| | | | Heavy drinking, Obesity (%), BCG vaccination (%) | Negative |
| Sohn (2019) [a] | Mortality | Spatial analysis with Poisson regression | Deprivation index | Positive |
| | | | SO2, Diabetes, O3, CO, heavy drinking | Not significant |
| Sun (2015) | Prevalence | Partial least square path modelling and GWR | Air quality | Mostly negative |
| | | | Climatic factors, education, primary industry employment, altitude | Entirely positive |
| | | | Economic level | Mostly positive |
| | | | Health service, unemployment level | Not significant |
| | | | Longitude factor and Population density | Negative effects |
| | | | NO2 (µg m−3), relative humidity (%) | Not significant |
| Wang (2019) | Prevalence | A spatial panel model | Proportion of male, average GDP, humidity, and wind speed | Not significant |
| | | | Rural residence, birth rate, air pressure, sunshine duration | Positive |
| | | | Number of beds, population density, precipitation | Negative |

*(Continued)*

**Table 1.** (Continued)

| First author (Year) | Primary outcome | Analytic method used | Ecological level factors investigated | Direction of association |
|---|---|---|---|---|
| Zhang (2019) | Incidence | GWR | Average rainfall | Positive |
| | | | Average humidity, average sunshine duration, average temperature, average wind speed and average air pressure | Negative |
| Wei (2016) | Incidence | GWR | Population density, proportion of minorities | Positive |
| | | | Proportion of agricultural population | Mostly positive |
| | | | Per capita gross domestic product (GDP) | Negative |
| Wubuli (2015) | Incidence | GWR | Proportion of minorities | Positive |
| | | | Per capita GDP | Negative |
| Chaw (2022) * | Incidence | Negative binomial model | Average temperature, average rainfall | Positive |
| | | | Average wind speed, relative humidity, and sunshine duration | Not significant |
| Chen (2016) * | Incidence | Logistic regression model | Particulate matter (PM10) (μg/m3) | Positive |
| | | | SO2, NO2, CO and O3 | Not significant |
| Huang (2020) * | Incidence | Distributed lag non-linear model (DLNM) | NO2 (10 μg/m3) | Positive |
| | | | SO2 (10 μg/m3) and O3 (10 μg/m3) | Negative |
| | | | PM2.5 (10 μg/m3) | Not significant |
| Hwang (2014) * | Incidence | Gaussian conditional autoregressive model | SO2 | Positive |
| | | | PM10, O3, CO, NO2 | Not significant |
| Jassal (2013) * | Prevalence | Logistic regression model | PM2.5 (mg/m3) | Positive |
| | | | O3 | Not significant |
| Kim (2020) * | Incidence | Generalized linear mixed | PM10 and SO2 | Positive |
| Kuddus (2019) * | Incidence | Generalized linear Poisson | Temperature, humidity, and rainfall | Positive |
| Lai (2016) * | Incidence | Cox proportional hazards regression | Nitrogen dioxide | Positive |
| | | | O3 | Negative |
| | | | Particulate matters (PM10 and PM2.5) | Not significant |
| Li (2023) * | Prevalence | Exposome-wide association study | PM10, PM2.5, NO2, CO, temperature, Precipitation, wind speed | Positive |
| | | | SO2 and altitude | Negative |
| | | | Relative humidity and O3 | Not significant |
| Peng (2017) | Mortality | Cox proportional hazards | Particulate matter (PM2.5) | Positive |
| Nie (2022) * | Incidence | Generalized additive model | Relative humidity | Positive |
| | | | Wind speed (m/s) | Negative |
| | | | Temperature (˚C) | Not significant |
| Smith (2016) * | Prevalence | Conditional logistic regression | Particulate matter (PM10) and nitrogen dioxide | Positive |
| | | | Particulate matter (PM2.5), SO2, O3 and CO | Not significant |
| Wang (2021) * | Incidence | Generalized linear model | Particulate matter (PM10) | Positive |
| | | | SO2 | Not significant |
| Zhu (2018) * | Incidence | Quasipoisson model | Sulphur dioxide and Nitrogen dioxide | Positive |
| | | | Particulate matter (PM10) | Not significant |
| Zhang (2022) | Incidence | System Generalized Method of Moments (System GMM) | GDP per capita, proportion of urban population, sunshine duration | Negative |
| | | | Proportion of the illiterate population | Positive |
| | | | PM10, average temperature and average relative humidity | Not significant |

**Note:** [a] single study with two outcomes

*Included in meta-analysis, GWR: geographically weighted regression; EDHS: Ethiopian Demographic and Health Survey; CAR: conditional autoregressive; INLA: Integrated Nested Laplace Approximation; GIS: geographic information system; OLS: Ordinary least squares regression; PLHIV: people living with HIV.

logistic regression, Poisson regression, generalized additive model and others (Table 1 and S2 Table).

## Ecological level factors associated with TB incidence

**Climatic and geographic factors.** Twenty studies, half of which originated from China, explored the impact of climatic factors on TB incidence. While findings exhibited some discrepancies, a prevailing trend suggests that increased mean temperature (10/18 studies), humidity (9/15), precipitation (4/6), rainfall (4/8), and elevation (1/2) were positively associated with TB incidence in at least half of studies examining these factors. However, this trend encountered challenges, as two provincial-level studies from China contradicted the association between TB incidence and mean temperature, and one study revealed a negative association with mean humidity, rainfall, and precipitation. Additionally, some studies yielded insignificant associations between mean temperature (6/18), precipitation (1/6), and rainfall (1/8) with TB incidence.

Conversely, factors such as average wind speed (5/11 studies), average air pressure (3/5), altitude (2/4), and slope (1/1) displayed predominantly negative associations with TB incidence across the studies. Sunshine duration displayed a slightly higher frequency of negative association (3/8) compared to positive ones (in 2 out of 8). Positive associations were also noted for air pressure (2/5 studies). Notably, only one county-level study in China, utilizing a five-year average of climatic data, reported a positive association for average wind speed, while no conflicting findings were observed for altitude and slope (Fig 2).

**Air quality measures.** Fourteen studies assessed the association between air quality-related factors and the incidence of TB yielding slightly consistent findings except for SO2 where significant conflicting results have been reported. The environmental factors including air pollution with sulphur dioxide ($SO_2$) (6/12), nitrogen dioxide ($NO_2$) (6/9), particulate matter (2.5-10($\mu g/m^3$)) (6/15), and carbon monoxide (CO) (2/6) had a positive association with TB incidence. A village-level study in Malaysia, with environmental data from different periods,

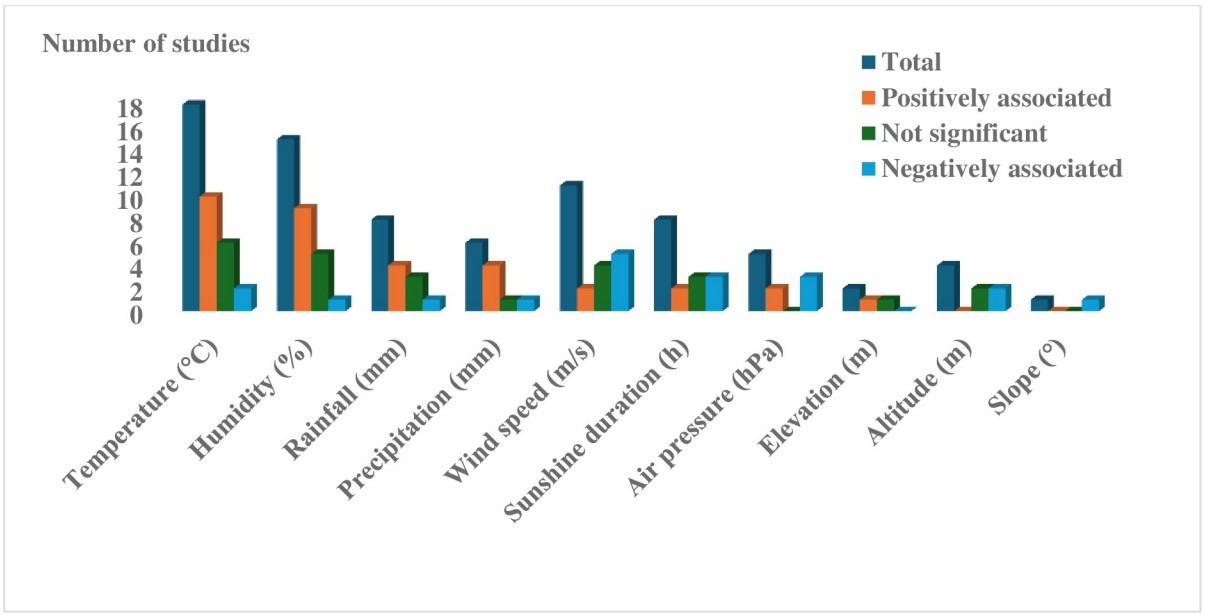

**Fig 2. Climatic and geographic factors associated with TB incidence.**

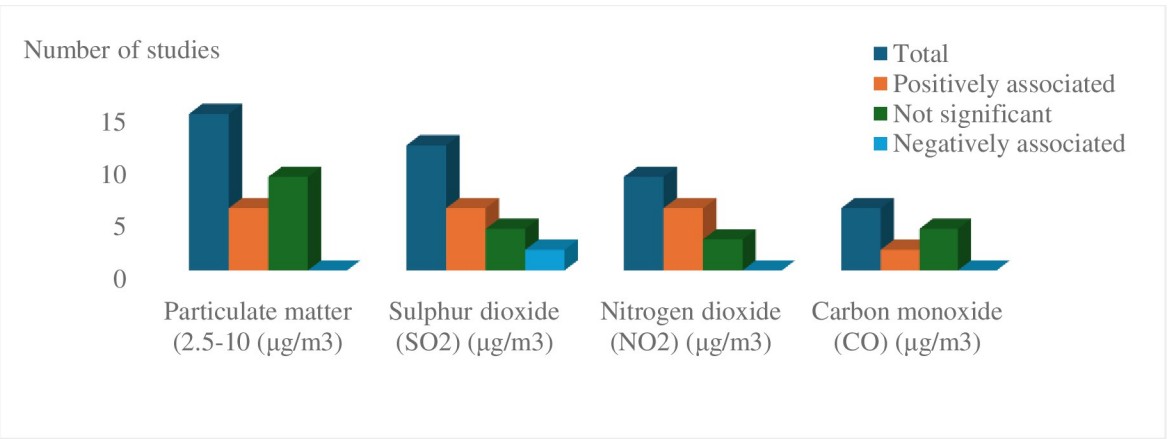

**Fig 3. Air quality-related factors associated with TB incidence.**

also showed that bad air quality containing high concentrations of $NO_2$, $SO_2$, CO, and inhalable particulate matter, was associated with increased incidence of TB by varying degrees at different geographic locations. In contrast, two studies from China reported a negative association of $SO_2$ air pollution with TB incidence. A negative association was also recorded between ozone ($O_3$) (2/8) concentration and TB incidence. However, several studies reported for a non-significant association between $SO_2$ (4/12), particulate matter (8/14), $NO_2$ (3/9), CO (4/6), O3 (6/8) and TB incidence (Fig 3).

**Socio-economic and demographic factors.** TB incidence was positively associated with several socio-economic and demographic factors including higher population density (10/14), higher internal migration (3/4), higher urbanization (3/8), higher proportion of males (2/4), higher illiteracy rate (2/5), higher unemployment rate (1/2), higher birth rate (1/2), and higher level of minority ethnic representation in the population (1/1). In contrast, good knowledge about TB (2/2), low population growth rate (1/1), and low level of social vulnerability (1/1), were reported to be negatively associated with TB incidence.

Common economic indicators such as increased total gross domestic product (GDP) (4/9 studies), per-capita GDP (2/4), and high wealth index (2/8) were negatively associated with TB incidence although positive associations with total GDP (1/8), per-capita GDP (1/4), and wealth index (2/8) were recorded in some instances. Economic inactivity (1/1) and poverty (4/4) were positively associated with TB incidence with no conflicting findings reported.

Health service-related factors such as higher health insurance participation rate (1/1), better access to health facilities (1/1), increased TB treatment success rates (2/2), and better BCG vaccination coverage (1/1) were reported to be protective factors for TB. However, a district-level ecological study in Ethiopia reported a positive association between access to health facilities and TB incidence.

TB incidence was also positively associated with the prevalence of co-morbidities including increased HIV/AIDS (1/2) and diabetes (1/1) prevalence, although one study has reported an opposite association for HIV/AIDS prevalence. Some ecological studies reported a non-significant association between TB incidence and travel time to the city (1/1), distance to health facilities (1/1), investment in TB control (2/2), healthcare funding (2/2), number of grass-roots health facilities (1/1), number of people per medical personnel (1/1), and proportion of people living with HIV ever enrolled in antiretroviral therapy (1/1) (Tables 1 and 2).

**Ecological factors associated with TB mortality.** Four studies (two from Brazil) have investigated the effect of ecological factors on TB mortality. TB mortality was positively

**Table 2. Aggregated population-level ecological factors associated with TB incidence.**

| Socio-economic and demographic factors | Number of studies (N) | Association with TB incidence |
|---|---|---|
| Gross domestic product (GDP) | **9** | Positively associated (n = 1) |
| | | Negatively associated (n = 4) |
| | | Not significant (n = 4) |
| Per capita GDP | 4 | Positively associated (n = 1) |
| | | Negatively associated (n = 2) |
| | | Not significant (n = 1) |
| Wealth index | 8 | Positively associated (n = 2) |
| | | Negatively associated (n = 2) |
| | | Not significant (n = 4) |
| Poverty level | 4 | Positively associated (n = 4) |
| | | Negatively associated (n = 0) |
| | | Not significant (n = 0) |
| Proportion of internal immigrants | **4** | Positively associated (n = 3) |
| | | Negatively associated (n = 0) |
| | | Not significant (n = 1) |
| Urban population (%) | **8** | Positively associated (n = 3) |
| | | Negatively associated (n = 3) |
| | | Not significant (n = 2) |
| Male population (%) | 4 | Positively associated (n = 2) |
| | | Negatively associated (n = 0) |
| | | Not significant (n = 2) |
| Population density (people/km$^2$) | 14 | Positively associated (n = 10) |
| | | Negatively associated (n = 2) |
| | | Not significant (n = 2) |
| Good knowledge about TB | 2 | Positively associated (n = 0) |
| | | Negatively associated (n = 2) |
| | | Not significant (n = 0) |
| Unemployment rate | 2 | Positively associated (n = 1) |
| | | Negatively associated (n = 0) |
| | | Not significant (n = 1) |
| Proportion of illiterate | 5 | Positively associated (n = 2) |
| | | Negatively associated (n = 0) |
| | | Not significant (n = 3) |
| Birth rate | 2 | Positively associated (n = 1) |
| | | Negatively associated (n = 0) |
| | | Not significant (n = 1) |
| Proportion elderly (%) | 1 | Positively associated (n = 1) |
| | | Negatively associated (n = 0) |
| | | Not significant (n = 0) |
| Proportion minority population | 1 | Positively associated (n = 1) |
| | | Negatively associated (n = 0) |
| | | Not significant (n = 0) |
| Population growth rate | 1 | Positively associated (n = 0) |
| | | Negatively associated (n = 1) |
| | | Not significant (n = 0) |

(*Continued*)

**Table 2.** (Continued)

| Socio-economic and demographic factors | Number of studies (N) | Association with TB incidence |
|---|---|---|
| Social vulnerability | 1 | Positively associated (n = 0) |
| | | Negatively associated (n = 1) |
| | | Not significant (n = 0) |
| BCG vaccination (%) | 1 | Positively associated (n = 0) |
| | | Negatively associated (n = 1) |
| | | Not significant (n = 0) |
| Health insurance participation rate | 1 | Positively associated (n = 0) |
| | | Negatively associated (n = 1) |
| | | Not significant (n = 0) |
| TB treatment success rate | 2 | Positively associated (n = 0) |
| | | Negatively associated (n = 2) |
| | | Not significant (n = 0) |
| Access to health facilities | 1 | Positively associated (n = 0) |
| | | Negatively associated (n = 1) |
| | | Not significant (n = 0) |
| HIV infection rate | 1 | Positively associated (n = 1) |
| | | Negatively associated (n = 0) |
| | | Not significant (n = 1) |
| Diabetes (%) | 1 | Positively associated (n = 1) |
| | | Negatively associated (n = 0) |
| | | Not significant (n = 0) |
| Proportion of economically inactive population | | Positively associated (n = 1) |
| | | Negatively associated (n = 0) |
| | | Not significant (n = 0) |

**Note: N:** total number of studies per factor; n: number of studies per direction of association: n/N information used to summarize the direction of association in narrative synthesis.

associated with higher population density (1/1), increased social deprivation index (1/1), and higher particulate matter concentration (2.5–10 (μg/m3) (1/1). In contrast, high household income (2/2) was negatively associated with TB mortality rates. However, associations between TB mortality and unemployment rate, and measures of environmental pollution, including sulphur dioxide, nitrogen dioxide, and carbon monoxide were non-significant (Fig 4).

## Meta-analysis results

A total of twenty-four studies that examined the association between ecological factors and TB incidence and reported compatible effect sizes were considered methodologically suitable for inclusion in the meta-analysis. Among these, several factors were found to be associated with an increased risk of TB incidence, including higher average relative humidity (%) (pooled adjusted relative risk (RR) = 1.45, 95%CI:1.12, 1.77, n = 6, I2 = 87.85%), higher average rainfall (mm) (RR = 1.56, 95%CI: 1.11, 2.02, n = 6, I2 = 84.74%), increased exposure to sulphur dioxide (μg m−3) (RR = 1.04, 95% CI:1.01, 1.08, n = 7, I2 = 92.19%), increased fine particulate matter concentration (PM2.5) (μg/ m$^3$) (RR = 1.33, 95% CI: 1.18, 1.49, n = 3, I2 = 0.00%), and higher population density (people/km2) (RR = 1.01,95%CI:1.01–1.02, n = 2, I2 = 0.00%). Conversely, higher average wind speed (m/s) (RR = 0.89, 95%CI: 0.82,0.96, n = 3, I2 = 97.81%) was associated with a decreased risk of TB incidence (Fig 5). Exposure to carbon monoxide (pooled

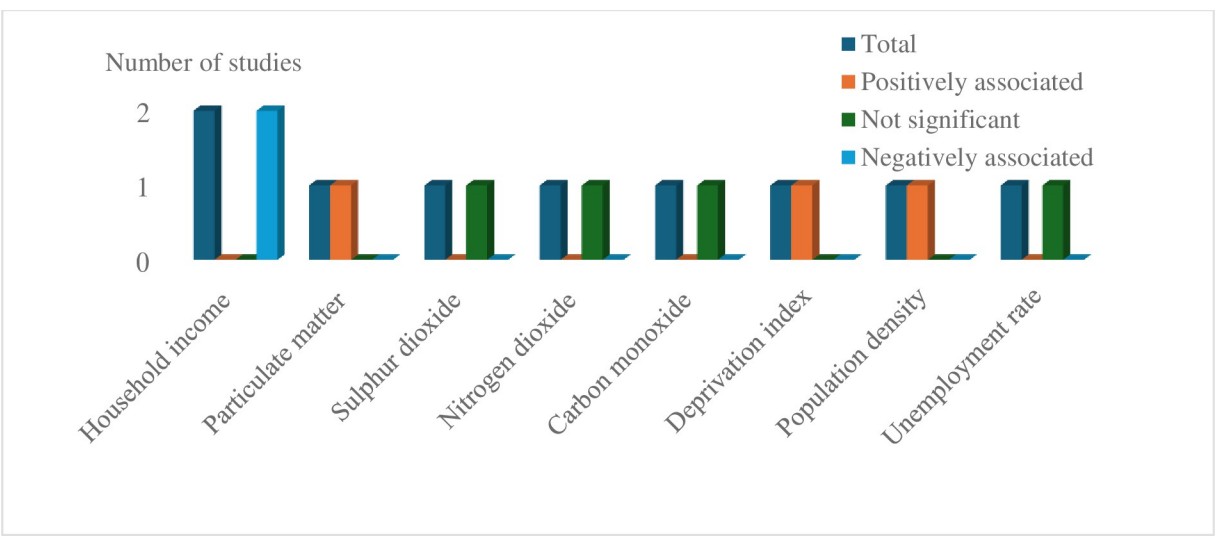

**Fig 4. Ecological factors associated with TB mortality.**

adjusted odds ratio (OR) = 1.25, 95% CI:1.21,1.29, n = 2, I2 = 0.00%) and coarse particulate matter (PM$_{10}$) (OR = 1.03, 95% CI 1.02,1.05, n = 2, I2 = 0.00%) were found to be associated with increased odds of TB prevalence (S3 Table and S1 and S2 Figs).

## Quality assessment

Overall, the quality scores ranged from 12 to 21, with mean scores in specific domains such as statistical methodologies used and reporting quality being 2.75 out of 3 and 5.67 out of 6

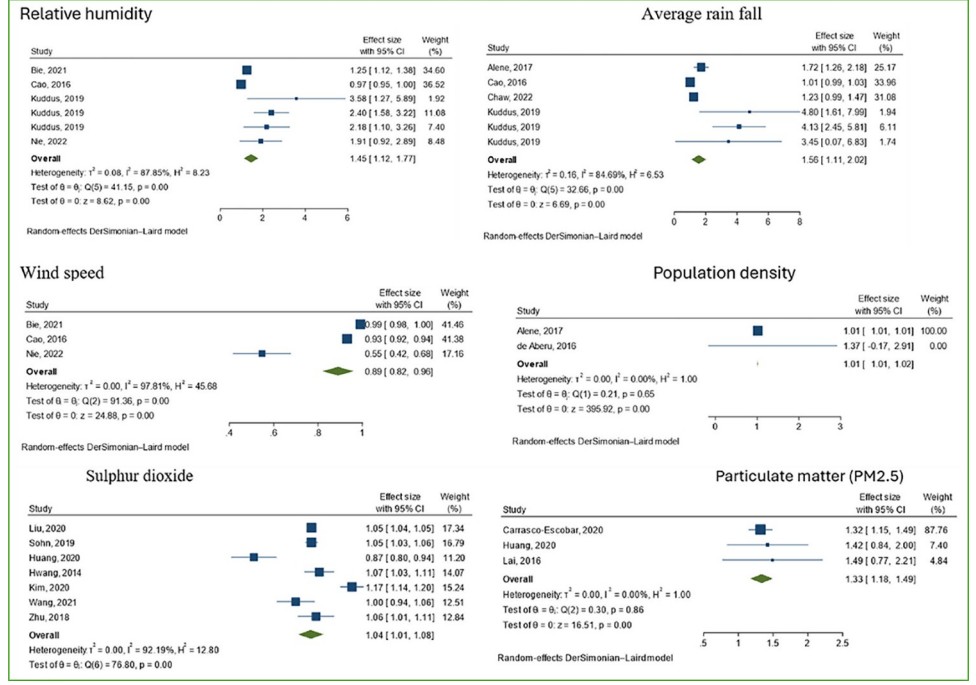

**Fig 5. Forest plot of pooled effects of ecological factors associated with TB incidence.**

respectively. The study design domain had an average score of 9.45 out of 12, with scores ranging from 7 to 11. The majority of studies included demonstrated generally high methodological quality, with most scoring well across various domains (S2 Table and S3 Fig). However, there were still some notable areas where biases could potentially influence the results. For instance, one common bias related to study design was the inclusion of all available data without estimating the sample size. This can affect the representativeness of the sample and consequently the validity of the inferences drawn from the study. The other common bias observed, particularly in most low-quality studies, was the tendency to pay insufficient attention to discussing the impact of some common biases in ecological studies including ecological fallacy on their findings. Detailed scores for each item across all domains for every study are available in the S1 File. A statistical investigation of publication bias using Egger's regression test, returned non-significant p-values indicating the absence of bias for studies that assessed sulphur dioxide, fine and coarse particulate matter, carbon monoxide, wind speed, and population density (S4 Table). Conversely, significant Egger's regression p-values in studies investigating relative humidity and average rainfall suggest publication bias (S4 Table).

No heterogeneity (I2 = 0.00%) was found among studies examining population density, particulate matter, and carbon monoxide. However, studies investigating climatic factors and sulphur dioxide exhibited high heterogeneity (Fig 5).

## Discussion

This study is the first comprehensive systematic review and meta-analysis that qualitatively explored and quantitatively analysed the effect of ecological-level factors on TB incidence and mortality. The findings indicate that TB incidence and mortality were associated with various ecologic factors including climatic, air quality, socioeconomic and health service-related factors.

Our narratively synthesised evidence showed that climatic factors including high average temperature, high average relative humidity, high average rainfall, and high precipitation were predominantly positively associated with TB incidence. The positive association between high average relative humidity and average rainfall and TB incidence was further supported by the current meta-analysis findings. The possible explanation for the positive effect of relative humidity on TB incidence might be the higher survival rate of *M. tuberculosis* (MTB) in an environment with high humidity [21]. The higher the humidity the slower the air circulation and this then creates favourable conditions for the transmission of tubercle bacilli [81]. On the other hand, in geographic locations with high humidity, there might be less exposure to sunlight, and this might, in turn, reduce vitamin D absorption [82]. A low level of vitamin D leads to impaired cellular immunity, which is an important risk factor that activates mycobacterial infections after a long time [81, 83]. The co-occurrence of this important climatic factor (i.e., relative humidity) with other weather-based factors including low wind speed may also facilitate the formation of large tubercle bacillus containing aerosols then increasing the infectious bacterial loads that may surpass the immune system defences and progresses to TB disease in susceptible human hosts [84]. High rainfall is also associated with an increased risk of TB incidence. In areas with heavy rainfall, there is often an increase in humidity and the potential for poor sanitation conditions, which can facilitate the proliferation of pathogens including MTB [85]. Additionally, high rainfall may lead to overcrowding as people seek shelter in close quarters, which can enhance the transmission of TB within population [86]. Conversely, high average wind speed is associated with a lower risk of TB, possibly because higher wind speeds drive better ventilation and lower pollutant concentrations thereby indirectly reducing the risk of pulmonary TB.

In the present study, we found that outdoor air pollution with particulate matter, carbon monoxide and sulphur dioxide was significantly associated with a higher incidence of TB. Biologically, exposure to air pollutants reduces resistance to infection because of damage to the airway system, epithelial permeability, and macrophage function, which potentially leads to easy progression of TB infection to overt TB disease [87]. Secondly, the continuous exposure to these pollutants reduces the production or release of tumour necrosis factor-α (TNF-α), interferon-γ (IFN-γ) and interleukin-1 which play an important role in combating MTB infection [88]. Moreover, TNF-α is a very crucial cytokine in host defences against *M. tuberculosis* via granuloma formation [89]. Therefore, the geographic areas with higher concentrations of these combustion-related air pollutants, mostly in urban settings, might be at greater risk of TB.

Another important ecological factor that positively affects the incidence of TB was population density. In our narrative synthesis, it has been most consistently reported as a risk factor for TB incidence and mortality. The pooled effect in the current meta-analysis also shows its positive association with TB incidence. Previous research supports this association as overcrowding increases the likelihood of TB infection and transmission within communities [90]. Moreover, population density can exacerbate existing social vulnerabilities including lack of sanitation, inadequate housing, crowded public transport, and overloaded health services, particularly in low socioeconomic settings and this could potentially lead to higher TB incidence. In contrast, although population density is commonly studied and consistently identified as a risk factor for TB, a county-level study in Mongolia, a high-TB-burden country [91], found a negative association with TB risk. Similarly, another study examining the local impact of social factors on TB distribution noted that population density had a negative influence on TB incidence across most included regions. This trend may be attributed in part to the process of urbanization, as Mongolia for instance has undergone rapid urban development [92]. Urban areas typically offer better access to healthcare facilities, diagnostic services, and treatment options compared to rural regions [93]. Consequently, densely populated urban centers may experience lower TB incidence rates due to enhanced healthcare accessibility and early case detection and treatment. This was consistent with the finding in the current synthesized evidence where three studies reported a negative association between increased urbanization and TB incidence.

Although there have been inconclusive findings for some factors reported in primary studies, our narrative synthesis documents a range of ecological factors that have played a significant role in TB risk. Negative socioeconomic indicators such as poverty, economic inactivity within a population, immigration, unemployment, urbanization, high proportions of aging and minority populations, high birth rate, illiteracy were reported as important TB risk factors. Conversely, positive socioeconomic indicators such as total gross domestic product (GDP), per-capita GDP and wealth indices, and health service-related factors including participation rate of health insurance, access to health facilities, the success rate of TB treatment, and proportion of BCG vaccination were negatively associated with TB incidence. This creates an overall picture of a negative association between prosperity and disease risk.

The current study has also synthesized the ecological factors associated with TB mortality. The results show that social deprivation and particulate matter were positively associated with TB mortality. In previous studies, social deprivation [94] and social inequalities were associated with high TB mortality rates which is consistent with the current results. This might be because the benefits of wealth for health depend on the distribution of wealth in the population, since a healthy society is not exclusive to a rich population, but rather to a population in which the income gap between rich and poor is narrow [95]. Therefore, this social relativity might contribute to increased TB mortality in addition to the absolute standards of living.

## Limitations

This study has some important limitations. First, although an adequate number of studies were included in the narrative synthesis, few studies were eligible for meta-analysis due to large variation in effect sizes reported and a difference in variable definitions used across studies, and this might have affected the direction of association during pooled effect size estimation. The second limitation is that we found higher heterogeneity among studies that investigated climatic factors. A similar finding was observed in a previous meta-analysis of climatic factors with another respiratory disease study [96]. Since few studies per factor were included, it was not feasible either to investigate the source of heterogeneity through subgroup analysis or to reduce the heterogeneity by conducting sensitivity analysis. Conducting a subgroup analysis by geographic regions would have been beneficial to account for differences in health service infrastructure, socioeconomic conditions, and climatic factors. This approach helps identify how TB risk factors may vary across different locations and provides insights into regional-specific influences on TB incidence. The third limitation is related with the narrative synthesis method employed. The vote-counting method in narrative synthesis of the direction of association that was used in the current study does not account for the strength or magnitude of the association. Instead, it simply tallies the number of studies reporting a positive, negative, or neutral association without considering the effect sizes or the quality of evidence. Finally, the association observed between TB incidence and some aggregate ecological factors may be prone to ecological fallacy i.e., a failure in epidemiological reasoning that happens when an inference is made for an individual based on aggregate data. Thus, ecological fallacy may obscure or misrepresent the true association of variables, and this might then lead to misleading conclusions. Findings grounded on aggregate data should be interpreted with caution.

## Conclusion

This systematic review indicates that TB incidence is significantly associated with a range of ecological drivers including average relative humidity, average rainfall, wind speed population density, and various combustion-related air pollutants. Moreover, different population-level, socioeconomic and health service-related factors were associated with TB incidence and mortality. Designing a comprehensive population-level framework beyond an individual-level approach is vital to mitigate and reduce TB incidence and mortality. Therefore, implementing a strong public health intervention that would improve air quality, and socioeconomic status and focusing on mitigation measures against the effects of climate could help to achieve the targets of end TB strategy and fasten progress to elimination.

## Supporting information

**S1 Table. Search strategies for ecological factors of TB incidence and mortality.**
(DOCX)

**S2 Table. Descriptive summary table for included studies.**
(DOCX)

**S3 Table. A summary of the pooled effect estimates for factors associated with TB incidence examined in studies reporting odds ratios.**
(DOCX)

**S4 Table. Egger's regression test results.**
(DOCX)

**S5 Table. Adapted quality assessment tool template and definitions.**
(DOCX)

**S6 Table. A numbered table of all studies identified in the literature search, with the full title of each article, including those that were excluded from the analyses.**
(DOCX)

**S1 Fig. Forest plot of pooled effect (odds ratio) of carbon monoxide on TB incidence.**
(DOCX)

**S2 Fig. Forest plot of pooled effect (odds ratio) of particulate matter (PM10) on TB incidence.**
(DOCX)

**S3 Fig. Quality of 52 included studies.**
(DOCX)

**S1 Data. A table of all data extracted from the primary research sources for the systematic review and/or meta-analysis.**
(XLSX)

**S1 File. Quality assessment scores for 42 studies across each item of the three domains.**
(XLSX)

## Author Contributions

**Conceptualization:** Alemneh Mekuriaw Liyew, Archie C. A. Clements, Beth Gilmour, Kefyalew Addis Alene.

**Data curation:** Alemneh Mekuriaw Liyew, Temesgen Yihunie Akalu, Beth Gilmour, Kefyalew Addis Alene.

**Formal analysis:** Alemneh Mekuriaw Liyew, Temesgen Yihunie Akalu.

**Funding acquisition:** Kefyalew Addis Alene.

**Investigation:** Kefyalew Addis Alene.

**Methodology:** Alemneh Mekuriaw Liyew, Archie C. A. Clements, Temesgen Yihunie Akalu, Beth Gilmour, Kefyalew Addis Alene.

**Software:** Alemneh Mekuriaw Liyew, Archie C. A. Clements, Beth Gilmour, Kefyalew Addis Alene.

**Supervision:** Archie C. A. Clements, Beth Gilmour, Kefyalew Addis Alene.

**Validation:** Alemneh Mekuriaw Liyew, Archie C. A. Clements, Temesgen Yihunie Akalu, Beth Gilmour, Kefyalew Addis Alene.

**Visualization:** Alemneh Mekuriaw Liyew, Archie C. A. Clements, Temesgen Yihunie Akalu, Beth Gilmour, Kefyalew Addis Alene.

**Writing – original draft:** Alemneh Mekuriaw Liyew.

**Writing – review & editing:** Alemneh Mekuriaw Liyew, Archie C. A. Clements, Temesgen Yihunie Akalu, Beth Gilmour, Kefyalew Addis Alene.

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
