## [Decision Letter · Decision Letter 0]

29 May 2024

PGPH-D-24-00826

Ecological-level factors associated with tuberculosis incidence and mortality: A systematic review and meta-analysis.

Dear Dr. Liyew,

Thank you for submitting your manuscript to PLOS Global Public Health. After careful consideration, we feel that it has merit but does not fully meet PLOS Global Public Health’s publication criteria as it currently stands. Therefore, we invite you to submit a revised version of the manuscript that addresses the points raised during the review process.

We look forward to receiving your revised manuscript.

Kind regards,

Patrick Musicha

Academic Editor

Journal Requirements:

2. Please provide separate figure files in .tif or .eps format only and remove any figures embedded in your manuscript file. Please also ensure all files are under our size limit of 10MB.

4. In the online submission form, you indicated that "Data will be available upon request from the corresponding author". 

3. Uploaded as supplementary information.

Additional Editor Comments (if provided):

Reviewers' comments:

Reviewer's Responses to Questions

**Comments to the Author**

1. Does this manuscript meet PLOS Global Public Health’s publication criteria? Is the manuscript technically sound, and do the data support the conclusions? The manuscript must describe methodologically and ethically rigorous research with conclusions that are appropriately drawn based on the data presented.

Reviewer #1: Yes

Reviewer #2: Yes

2. Has the statistical analysis been performed appropriately and rigorously?

Reviewer #1: I don't know

Reviewer #2: Yes

3. Have the authors made all data underlying the findings in their manuscript fully available (please refer to the Data Availability Statement at the start of the manuscript PDF file)?

Reviewer #1: No

Reviewer #2: Yes

4. Is the manuscript presented in an intelligible fashion and written in standard English?

Reviewer #1: Yes

Reviewer #2: Yes

5. Review Comments to the Author

Reviewer #1: This is a generally well-written article that has been clearly justified. Please see suggestions below to improve clarity and reporting.

Introduction:

1. I find the use of the term ecological factors slightly confusing as it is often associated with environmental factors. I think it would be useful for authors to clarify what they define as ‘ecological factors’ or consider using a more encompassing term such as ‘population-level factors’.

Methods:

1. Author stated in protocol that Newcastle Ottawa would be used to assess bias. Please justify the reason for using a different risk of bias tool and transparently report differences between protocol and review.

2. Unclear why ref number 37 is included in the risk of bias section.

3. “Narrative synthesis was used to qualitatively summarize the effect of ecological factors on the primary outcome of interest including TB incidence and mortality” This isn’t very clear – please could the authors provide more detail regarding the methods to summarise this data, see SWiM guidance https://doi.org/10.1136/bmj.l6890

Results:

1. Suggest streamlining table 1 as it is too large at present. I would include column 1, 3, 6 and 8 here. Risk of bias/quality results should be presented in a separate table with the narrative explanation of judgements. Other data could be reported in an Appendix.

2. The authors need to include denominators in their results – currently included sentences such as “TB incidence was consistently positively associated with mean temperature (n=10)”, but it is unclear whether only 10 studies measured this and all reported this association (in which case report as 10/10 studies) or whether 20 studies reported this outcome but the positive association was only seen in 10 studies (then report as 10/20 studies). This needs to be done throughout the results.

3. The authors have stated sentences such as “TB incidence was consistently positively associated with mean temperature (n=10), average humidity (n=9), precipitation (n=4), rainfall (n=4) and elevation (n=1).”. Do the mean that higher mean temperatures or lower mean temperatures are positively associated with TB incidence? Higher or lower precipitation? Etc. This needs to be clarified throughout the results.

4. For studies that report a conflicting result, such as the study in Mongolia that reported a negative association between TB and population density, are authors able to offer any explanation for this result?

5. Add denominators to table 2 and 3 to show total number of studies reporting each outcome.

6. I wonder if there is a more visual way to present the data in table 2 and 3, maybe a stacked bar chart?

Meta-analysis results:

1. Please explain definition of “methodologically suitable for inclusion in meta-analysis”.

2. Again, it is not clear whether increased or decreased humidity, rainfall, population density, etc are associated with increased risk of outcomes. Please clarify as this is important for the reader to understand and interpret the results.

3. Please narratively acknowledge the high I2 value associated with some results and discuss the implications of this.

4. In Table 4 please provide footnote explaining that results in bold represent significant results.

Quality assessment:

1. Please could authors provide a separate table here with results of risk of bias assessment and provide specific examples of issues with studies that resulted in low risk of bias scores. This allows the reader to understand common issues with these studies and aides’ interpretation of findings.

Discussion:

1. Suggest to add sub-headings for summary of evidence and limitations.

References:

1. Suggest separating included studies from other references if possible.

Supplementary files

1. Is it possible to add participant numbers, number of events, and risk of bias results to forest plots? See recent Cochrane reviews as an example.

2. Not recommended to conduct a funnel plot with fewer than 10 studies. Please consider whether these are reliable or useful.

3. Excluded studies – please be consistent with reason for exclusion such as using ‘irrelevant study design’ rather than ‘meta-analysis’

Other:

1. Please make all data publicly available either in supporting information or via an online repository.

Reviewer #2: The authors have produced a manuscript describing the ecological factors associated with TB incidence and mortality through the conduct of a systematic review and meta-analysis. The article is timely, and attempts to narrow the gap in knowledge of factors associated with TB epidemiology. The authors should try to address the following:

- The introduction fails to establish a strong rationale for the conduct of the study. Despite some inconsistencies in the literature about specific ecological factors associated with TB incidence and mortality, some consensus exists amongst TB researchers about these factors. There is thus a need to clearly outline why a comprehensive review on these factors is necessary to end the TB epidemic in line with End TB Strategy goals.

- The methods do not clearly define the "ecological factors" of interest that determined the papers that were to be included in this review. A statement about the population of interest (adult versus paediatric populations) would also be useful in fulfilling the PICOS requirements. A look at Table 2 which summarizes the ecological factors extracted from the included studies alludes to socioeconomic factors such as poverty indicators, population density and HIV testing practices being included as ecological factors. Using this table as the general definition of ecological factors, a quick search on Google Scholar brings up papers by Qianyun Zhang et al. (study from China published in Frontiers in Public Health in 2021), and Soares P et al. (published in Public Health in 2022) among others which are not included in the manuscript, suggesting that the search strategy was inadequate, which may be a result of poorly defining the Population of interest (P), Intervention/Exposures (I) (i.e. the ecological factors of interest), Comparators (C), and the Outcomes (O) (i.e. the measures of disease occurrence of interest).

- The authors produced forest plots for the meta-analysis outputs of individual ecological factors. It is understood that the journal has guidelines limiting the number of figures that may be included in the main manuscript. However, the forest plots are direct analytical outputs of a main objective the paper, and as such should feature in the main paper, and not the supplementary materials. The authors could group the figures using panels to create one illustration with a good figure legend in order to facilitate the inclusion of these in the main manuscript.

- Under the data extraction section on page 5, line 19: "analytyc" should be corrected to "analytic"

- In outlining potential limitations of the study, the ecological fallacy was rightly mentioned. However, the authors fail to suggest what effects the presence of the ecological fallacy would potentially have on the results of the study, merely stating it could result in "misleading conclusions".

- Figure S22: Quality of 52 included studies: provide units/axis title for the y (number) axis at the bottom

6. PLOS authors have the option to publish the peer review history of their article (what does this mean?). If published, this will include your full peer review and any attached files.

**Do you want your identity to be public for this peer review?** For information about this choice, including consent withdrawal, please see our Privacy Policy.

Reviewer #1: No

Reviewer #2: No

---

## [Decision Letter · Decision Letter 1]

2 Aug 2024

PGPH-D-24-00826R1

Ecological-level factors associated with tuberculosis incidence and mortality: A systematic review and meta-analysis.

Dear Dr. Liyew,

Thank you for submitting your manuscript to PLOS Global Public Health. After careful consideration, we feel that it has merit but does not fully meet PLOS Global Public Health’s publication criteria as it currently stands. Therefore, we invite you to submit a revised version of the manuscript that addresses the points raised during the review process.

While the reviewers' previous comments have been addressed, the paper can be reorganised to improve readability and additional justification can be provided for some points: 

Figure 1 is a useful schematic; however, for readers to be able to follow with the text, it would be good to use the same headings for subsection titles under “Methods”. E.g. the subsection “Data source and search strategy” up to line 120 of p.9 could be combined under a subsection titled “Identification”, lines 121 – 132 could be under a subsection titled “Screening” etc.Figure 1: How do the items (“Full-text articles excluded”, “Descriptive studies” etc.) relate to the items in the quality assessment tool (Table S8)?“Quality Assessment”, p.5: Please put this section after “Statistical analysis” to be in line with how the results are presented. Lines 172-174 on Egger’s regression test on publication bias can also be added to this section.Lines 168-174: It would be good to add a line or two to support the use of random effects meta-analysis here. You can reference this paper:

Riley RD, Higgins JP, Deeks JJ. Interpretation of random effects meta-analyses. BMJ2011;342:d549. doi:10.1136/bmj.d549. pmid:21310794

It mentions that “heterogeneity in treatment effects is caused by differences in study populations (such as age of patients), interventions received (such as dose of drug), follow-up length, and other factors”. In your analysis, the different spatial scales (e.g. county-level, provincial-level) of the studies could be an important factor influencing the treatment effects.

Please also note that the pooled result from a random-effects meta-analysis should be interpreted as the average of the intervention effects across studies and not an estimate of the common effect.

Line 175, p.6: “Results” can be a new section, containing the subsections “Characteristics of included studies”, “Ecological level factors associated with TB incidence”, “Ecological factors associated with TB mortality”. So, the main sections in the paper would be “Introduction”, “Methods”, “Results” and “Discussion”.Line 178, p.6: It seems that the total number of references here is 52 instead of 53.Table 1: Please be consistent with the terms used. E.g. use “Negative” instead of “Protective”, and “Positive” instead of “Risk”. Could you also indicate which of these rows were used for the meta-analysis?p.12, “Ecological level factors associated with TB incidence”: It may be better to present the narrative synthesis results before the meta-analysis results, to be in line with the flow of the "Methods" section. You can also draw links between the two results to provide credibility to your meta-analysis results. E.g. The pooled adjusted relative risk for average relative humidity is greater than 1 which is in line with the mainly positive association seen from the narrative synthesis.Line 240, p.13: The phrase “increased the occurrence of TB” seems to imply causality; however, I think only association was established. Please rephrase this.Lines 290-296, p.16: Will you be sharing the individual item scores for each paper (e.g. in a supplementary Excel file)? This will enable the readers can better understand the common biases mentioned.P.16-17, “Discussion”: The second paragraph on relative humidity and rainfall is confusing. Based on the narrative synthesis, high average relative humidity and high average rainfall are consistently associated with increased TB incidence. This is in line with the meta-analysis results. However, in lines 324-325, you mention that “Low rainfall is also associated with an increased risk of TB incidence”. Are you mentioning an exception to the rule? Is this context/location-dependent, and possibly related to the publication bias you mentioned on line 300?Lines 387-388: Please provide a reference for this respiratory disease study.Lines 388-390: Could you provide examples of the kinds of subgroup analyses you would consider if you had more studies? E.g. identifying similar characteristics in the locations associated with positive/negative effects and identifying the combinations of covariates used in the models (i.e. exploring potentially confounding factors) may be useful.There were several instances where “Poisson” was misspelt: on line 141 of p.5, line 194 of p.7 in Table S8 and multiple times in Table 1. Please correct these.

We look forward to receiving your revised manuscript.

Kind regards,

Michele Nguyen

Academic Editor

Journal Requirements:

Additional Editor Comments (if provided):

Reviewers' comments:

Reviewer's Responses to Questions

**Comments to the Author**

1. If the authors have adequately addressed your comments raised in a previous round of review and you feel that this manuscript is now acceptable for publication, you may indicate that here to bypass the “Comments to the Author” section, enter your conflict of interest statement in the “Confidential to Editor” section, and submit your "Accept" recommendation.

Reviewer #2: All comments have been addressed

2. Does this manuscript meet PLOS Global Public Health’s publication criteria? Is the manuscript technically sound, and do the data support the conclusions? The manuscript must describe methodologically and ethically rigorous research with conclusions that are appropriately drawn based on the data presented.

Reviewer #2: Yes

3. Has the statistical analysis been performed appropriately and rigorously?

Reviewer #2: Yes

4. Have the authors made all data underlying the findings in their manuscript fully available (please refer to the Data Availability Statement at the start of the manuscript PDF file)?

Reviewer #2: Yes

5. Is the manuscript presented in an intelligible fashion and written in standard English?

Reviewer #2: Yes

6. Review Comments to the Author

Reviewer #2: Comments have been addressed

7. PLOS authors have the option to publish the peer review history of their article (what does this mean?). If published, this will include your full peer review and any attached files.

**Do you want your identity to be public for this peer review?** For information about this choice, including consent withdrawal, please see our Privacy Policy.

Reviewer #2: No

---

## [Editor Report · Decision Letter 2]

29 Aug 2024

Ecological-level factors associated with tuberculosis incidence and mortality: A systematic review and meta-analysis.

PGPH-D-24-00826R2

Dear Mr Liyew,

We are pleased to inform you that your manuscript 'Ecological-level factors associated with tuberculosis incidence and mortality: A systematic review and meta-analysis.' has been provisionally accepted for publication in PLOS Global Public Health.

Best regards,

Michele Nguyen

Academic Editor